# Autophagy Dynamics and Modulation in a Rat Model of Renal Ischemia-Reperfusion Injury

**DOI:** 10.3390/ijms21197185

**Published:** 2020-09-29

**Authors:** Jean-Paul Decuypere, Shawn Hutchinson, Diethard Monbaliu, Wim Martinet, Jacques Pirenne, Ina Jochmans

**Affiliations:** 1Laboratory of Abdominal Transplantation, Transplantation Research Group, Department of Microbiology and Immunology, KU Leuven, B-3000 Leuven, Belgium; jeanpaul.decuypere@kuleuven.be (J.-P.D.); shawn.hutchinson@uhn.ca (S.H.); diethard.monbaliu@uzleuven.be (D.M.); jacques.pirenne@uzleuven.be (J.P.); 2Department of Abdominal Transplant Surgery, University Hospitals Leuven, B-3000 Leuven, Belgium; 3Department of Pharmaceutical Sciences, University of Antwerp, B-2610 Antwerp, Belgium; wim.martinet@uantwerpen.be

**Keywords:** ischemia-reperfusion injury, acute kidney injury, autophagy, apoptosis, trehalose

## Abstract

Renal ischemia-reperfusion (IR) injury leading to cell death is a major cause of acute kidney injury, contributing to morbidity and mortality. Autophagy counteracts cell death by removing damaged macromolecules and organelles, making it an interesting anchor point for treatment strategies. However, autophagy is also suggested to enhance cell death when the ischemic burden is too strong. To investigate whether the role of autophagy depends on the severity of ischemic stress, we analyzed the dynamics of autophagy and apoptosis in an IR rat model with mild (45 min) or severe (60 min) renal ischemia. Following mild IR, renal injury was associated with reduced autophagy, enhanced mammalian target of rapamycin (mTOR) activity, and apoptosis. Severe IR, on the other hand, was associated with a higher autophagic activity, independent of mTOR, and without affecting apoptosis. Autophagy stimulation by trehalose injected 24 and 48 h prior to onset of severe ischemia did not reduce renal injury markers nor function, but reduced apoptosis and restored tubular dilation 7 days post reperfusion. This suggests that trehalose-dependent autophagy stimulation enhances tissue repair following an IR injury. Our data show that autophagy dynamics are strongly dependent on the severity of IR and that trehalose shows the potential to trigger autophagy-dependent repair processes following renal IR injury.

## 1. Introduction

Renal ischemia-reperfusion (IR) injury is a major contributor to acute kidney injury (AKI), leading to acute tubular necrosis [1,2]. AKI is a very common condition, affecting 3–18% of hospitalized patients and 33–66% of those admitted to intensive care [3]. AKI is an independent risk factor for death, especially when renal replacement therapy is needed, and is associated with a mortality of 40–70% in critically ill patients [4,5]. Currently, there are no effective treatment strategies for AKI, and measures are supportive while recovery is awaited. Indeed, the kidney’s capacity to recover after ischemic injury is remarkable but not perfect [6,7]. In mild injury, this repair process restores renal structure and function; however, when the injury is severe, the repair process can trigger fibrosis, increasing the risk of developing chronic kidney disease [8]. Understanding the link between the severity of injury and the regeneration process could lead to the development of treatments that enhance recovery. A possible link is autophagy, an evolutionary conserved intracellular degradation pathway with homeostatic and damage-mitigating functions.

Macroautophagy, the best studied type of autophagy (and hereafter simply referred to as “autophagy”), manifests as intracellular vesicles (autophagosomes) that envelop cytoplasmic material and subsequently transport and deliver their cargo to the lysosomes for degradation. It involves more than 30 autophagy (Atg) proteins for the initiation, formation, transportation, and fusion of autophagosomes towards lysosomes [9]. Its levels depend on input from various signaling pathways, including nutrient signaling through the mammalian target of rapamycin (mTOR) and energy status monitoring through adenosine monophosphate -activated kinase AMPK. Typically, autophagy is stimulated upon stress as a survival pathway. This way, autophagy recycles damaged and toxic cytoplasmic material into cellular building blocks, which are then used to support anti-stress responses and energy maintenance. Moreover, autophagy is able to degrade damaged mitochondria, preventing the initiation of apoptosis, and therefore might reduce IR injury [10,11]. However, in addition to necrosis and apoptosis [12,13,14], autophagy has also been positively associated with injury during renal IR [15,16,17,18,19]. Indeed, autophagy can both prevent and assist cell death, depending on the type and duration of the stress, due to the molecular crosstalks between autophagy and cell death mechanisms such as apoptosis and necrosis (reviewed in [20,21]). This autophagy paradox—damaging on the one hand, protecting on the other—implies that the role and dynamics of autophagy during renal IR injury are not well understood [22]. We hypothesized earlier that the role of autophagy in renal IR injury depends on the duration of ischemia, where autophagy can switch from a protective to an injuring mechanism with an increasing ischemia time [23,24]. 

We now examined this hypothesis by evaluating the dynamics of autophagy, apoptosis, and injury in a rat renal IR model. Subsequently, we assessed the effects of autophagy stimulation on renal IR injury by the administration of trehalose, a naturally occurring disaccharide known to stimulate autophagy [25].

## 2. Results

### 2.1. Transient Renal Injury Following Mild Ischemia

All Sham and I45 rats (subjected to 45 min of mild ischemia) survived 90 days of reperfusion (R90d). The plasma creatinine was higher after I45 compared to the Sham rats up to R7d (Figure 1A), as were plasmatic aspartate aminotransferase (AST) and heart-fatty acid binding protein (h-FABP), especially early post reperfusion (Figure 1B,C). Terminal deoxynucleotidyl transferase end labeling (TUNEL) staining gradually increased with increasing reperfusion time, with the strongest signal at R6h (Appendix A). The positive area originated around the blood vessels (R1h) and migrated towards the tubuli at R6h and R24h. Cortical areas were most prominently stained, although some disperse staining was observed in the medulla. It should be noted that besides the expected TUNEL staining of the nuclei, we observed most prominently the staining of the tubular lumen, as previously described [26,27]. TEM revealed swollen and damaged mitochondria at R3h (Appendix A) in the kidney, confirming the occurrence of intracellular damage. TEM also revealed possible early and late autophagosomes, but clear differences between the Sham and I45 group were not observed. In addition, the mRNA transcription of inflammatory factors intracellular adhesion molecule-1 (ICAM-1) (Figure 1D), interleukins IL-6 and IL-10 (Appendix A) and stress marker heat shock protein 70 (Hsp70) (Figure 1E) significantly increased, with a peak around R3h. These results thus indicate that the kidneys are transiently injured in the I45 model, concomitant with decreased function, enhanced inflammation, and with sustained survival.

### 2.2. Autophagy Is Suppressed during Ischemia and Reperfusion Following Mild Ischemia

This rat IR model was now exploited to investigate the autophagic response to mild ischemia. Autophagy markers were assessed by Western blotting on Sham and I45 rat kidney lysates at several time points during reperfusion (R0h, R1h, R3h, R6h, R24h, R48h, R7d, and R90d) (Appendix A). During autophagy, LC3-I is converted into LC3-II, which then recruits to the autophagosomal membrane. As such, the LC3-II levels represent the amount of autophagosomes present at the time of kidney collection. As recommended in the literature, we quantified the ratio of LC3-II over the housekeeping protein Glyceraldehyde 3-phosphate dehydrogenase (GAPDH) [28]. Interestingly, ischemia alone (R0h) led to a reduction in LC3-II (Figure 2A). Since LC3-II eventually is degraded in the lysosomes, a decrease in LC3-II can signify reduced autophagosome formation or enhanced autophagosome clearance [28]. To establish the overall results of IR for the autophagic degradation rate, we also assessed the levels of Sequestosome 1 (Sqstm1/p62), a substrate for autophagic degradation. Ischemia alone (R0h) resulted in an elevation of p62, concomitant with the decrease in LC3-II (Figure 2B), suggesting suppressed autophagy. Since mTOR is the canonical signaling kinase negatively regulating autophagy, we evaluated the phosphorylation of ribosomal protein S6, which is indirectly phosphorylated by mTOR via the activation of p70S6 kinase. Reduced S6 phosphorylation was observed at R0h (Figure 2C), suggesting that the suppression of autophagy during ischemia could be mTOR-independent. 

Following mild ischemia, a decrease in LC3-II was generally detected compared to the Sham group, with significant reductions at R0h, R3h, and R24h (Figure 2A). Corresponding with a decrease in LC3-II, the p62 levels were significantly upregulated post reperfusion (Figure 2B), indicating that autophagy is suppressed during reperfusion in this model. Although phosphorylated S6 was decreased during ischemia (R0h), it was strongly increased post reperfusion (Figure 2C), suggesting that the decrease in autophagy could be partially explained by the negative regulation of autophagy by mTOR. It should be noted, however, that S6 can be phosphorylated by other kinases as well [29].

Next, we analyzed whether the results observed on the protein level were also reflected in the mRNA expression of LC3, p62, and BECN1. Interestingly, the LC3 and BECN1 expression were significantly reduced between R3h and R24h (Figure 2D,F). The sqstm1/p62 mRNA levels remained relatively stable, despite an increase at R6h, followed by a decrease at R24h (Figure 2E). As such, it is unlikely that the increase in the p62 levels on a protein level have a transcriptional cause, and most likely reflect the reduction in the autophagic degradation of the p62 protein. In conclusion, these data suggest that autophagy is most prominently suppressed during reperfusion in rats subjected to 45 min of ischemia. 

### 2.3. Apoptosis Is Enhanced during Reperfusion Following Mild Ischemia

We further investigated the apoptotic pathway post reperfusion. Therefore, Western blotting was performed for pro- (Bax, cleaved caspase 3) and anti-apoptotic (Bcl-2) markers on kidney protein lysates from Sham and I45 rats (Appendix A). Despite the reduction in autophagy, apoptosis was not affected following ischemia alone (R0h) (Figure 3A–C). While the Bax protein levels were significantly elevated between R1h and R7d (Figure 3A), the Bcl-2 levels remained stable (Figure 3B). Despite the fast increase in Bax, the levels of cleaved Caspase 3 did not increase before R48h (Figure 3C). Interestingly, a similar trend was observed for the mRNA expression of Bim, a pro-apoptotic BH3-only protein (Figure 3F). Unlike the protein levels, the Bax mRNA expression decreased in I45 rats at R24h and increased at R48h and R90d (Figure 3D). The expression of anti-apoptotic Bcl-2 mRNA was reduced at most time points post reperfusion in I45 rats (Figure 3E). Taken together, these results indicate that the apoptotic machinery is activated late post reperfusion (after 24 h) following mild ischemia in rat kidneys.

### 2.4. More Severe Ischemia Increases Kidney Damage

To understand whether autophagy is protective or detrimental for renal IR injury, we first compared the autophagic response in rats subjected to mild (I45) and severe (I60) ischemia. First, in the I60 group, 4/6 rats died within seven days following IR, while in both the I45 and Sham group, all the rats survived. Moreover, the plasma creatinine and AST were also higher in I60 compared to I45 at R3h and R24h (Figure 4A,B), while the plasma h-FABP was only significantly different at R3h (Figure 4C). In addition, the mRNA expression of inflammatory and stress markers ICAM-1 and Hsp70 were higher in I60 versus I45 at R3h (Figure 4D,E). The expressions of IL-6 and -10, however, were lower in I60 compared to I45 (Appendix A). Altogether, the I60 rat model inflicted more kidney damage and inflammation compared to the I45 model.

### 2.5. More Autophagy upon Severe Compared to Mild Ischemia

In the I60 model of severe ischemia, we then explored autophagy and apoptosis markers in the kidney. The results observed in the I60 model at the post-reperfusion times R0h, R3h, and R24h (Appendix A) were compared with these time points in the mild ischemia (I45) model. LC3-II decreased in both I45 and I60 in a similar trend, but its levels at R24h were significantly higher in I60 versus I45 (Figure 5A). Interestingly, this was associated with lower p62 levels in the I60 model (Figure 5B and Appendix A), suggesting a higher autophagic activity in I60 compared to I45. The phosphorylated S6 levels did not significantly differ between I45 and I60 (Figure 5C). On the mRNA level, the LC3 expression was, similarly to the protein levels, increased in I60 versus I45 (Figure 5D), while the p62 levels remained unaltered between the two groups (Figure 5E). The BECN1 expression was significantly lower in I60 versus I45 at R0h, but remained similarly downregulated in both groups afterwards (Figure 5F). Despite these changes in autophagy, no clear trend was observed in the apoptosis markers comparing I60 and I45 (Appendix A). These data thus indicate elevated autophagy upon severe ischemia compared to mild ischemia. 

### 2.6. Trehalose Stimulates Autophagy and Reduces IR Injury in the Kidney

To evaluate whether autophagy modulation would alter IR injury, rats were given 2 g/kg body weight trehalose by intraperitoneal injection 48 and 24 h prior to 60 min of ischemia, followed by 24 h or 7 days of reperfusion. Trehalose is a naturally occurring sugar synthesized by bacteria, fungi, and invertebrates, consisting of two glucose molecules connected by a 1-1 α bond. It is an mTOR-independent autophagy inducer that acts partially by inhibiting glucose transport [30]. The kidneys of trehalose-treated rats displayed higher LC3-I and LC3-II levels compared to vehicle-treated rats following 60 min of ischemia and 24 h of reperfusion, without affecting phosphorylated S6, suggesting no change in the mTOR activity and apoptosis (Figure 6A,B). Despite an improvement in survival after R7d (vehicle 50%; trehalose 83%), trehalose did not reduce renal injury at R24h, as evidenced by the similar TUNEL staining (Figure 6C) and plasma AST levels (Figure 6D). The mRNA expression of ICAM-1, Hsp70, and IL-10 even displayed an increasing trend upon trehalose treatment (Appendix A). In addition, the plasma creatinine was increased in the trehalose-treated rats (Figure 6E), suggesting reduced kidney function. However, at R7d, the trehalose-treated rat kidneys showed reduced TUNEL staining (Figure 6F) and an overall improved kidney structure with less dilated tubules (Figure 6G). Together, this suggests that trehalose-induced autophagy does not reduce AKI but improves the repair of the affected tissue.

## 3. Discussion

Renal IR injury is a common clinical complication and is the leading cause of AKI. With no effective treatment available to alleviate renal IR injury, it may progress into acute renal failure and increase mortality rates. Therefore, understanding the underlying mechanisms of this injury is crucial to reveal new therapeutic approaches. A central pathway in the progress of renal IR injury is autophagy, an evolutionary conserved intracellular catabolic pathway regulating cellular homeostasis and survival during stress. However, the exact role of autophagy in renal IR injury is still undetermined, as both protective and detrimental effects have been described [23,24]. Indeed, in certain conditions autophagy-dependent cell death can occur. This is suggested to be dependent on the dynamics of autophagy, where excessive or uncontrolled autophagy could trigger the initiation of cell death [19]. The role of autophagy in renal IR injury could therefore be dependent on the ischemic duration of the model, with mild ischemia triggering moderate, protective autophagy and severe ischemia triggering excessive, detrimental autophagy [23,24]. Here, we observed that autophagy dynamics are indeed strongly influenced by these parameters. 

First, autophagy was attenuated very rapidly following ischemia (Figure 2). This was not expected, as the lack of oxygen and subsequently energy reduction during ischemia should activate autophagy through AMPK [31]. However, AMPK-dependent autophagy activation occurs upon subtle changes in ATP production, while warm ischemia in rat kidneys leads to a fast and sudden fall in the ATP levels [32]. In view of the ATP dependency of the conjugation reactions leading to the autophagy-specific lipidation of LC3-I into LC3-II [33], this sudden lack of ATP is possibly the cause of the observed decrease in autophagy during ischemia. Similar observations of reduced LC3-II levels have also been observed following renal ischemia in mice [34]. 

Second, autophagy fluctuated during reperfusion (Figure 2), but overall the attenuation of autophagy during ischemia continues during reperfusion. This effect post reperfusion is most likely due to increased mTOR activity, in which the re-introduction of nutrients during reperfusion likely stimulates mTOR after a suppression caused by nutrient absence during ischemia [35]. This is reflected in the levels of phosphorylated S6, an indirect target of mTOR (Figure 2C). However, it should be noted that the phosphorylation status of S6 is the sum of multiple kinase activities (including mTOR and protein kinase A) and the activity of protein phosphatase-1 [29]. Ideally, the phosphorylation of ULK-1 at Ser757 should be assessed to determine the link between mTOR activity and autophagy [36]. 

Despite the dynamics of injury and autophagy, apoptosis occurred rather late post reperfusion (R48h). Although TUNEL staining was mostly observed early post reperfusion (R3–6h), this did not correspond with the levels of cleaved caspase 3. In addition, this staining was not typically nucleus-specific, as has been observed previously [26,27]. This suggests that TUNEL staining is indicative of apoptosis-independent DNA damage during renal IR. Indeed, massive reactive oxygen species production has been suggested to induce DNA damage independent of necrosis and apoptosis [37]. These data, therefore, suggest that that the initial (acute) injury post reperfusion is likely more associated with other (inflammatory) types of cell death (e.g., necrosis, caspase3-independent pyrroptosis, ferroptosis, or necroptosis) rather than apoptosis [38,39]. Indeed, in previous work we have shown that necroptosis inhibitor Nec-1 reduces the positive TUNEL staining 3 h post reperfusion in our model [26], suggesting that necroptosis is activated shortly following reperfusion. However, since Nec-1 did not affect the injury markers AST and h-FABP and the plasma creatinine levels, other types of cell death are likely to be activated as well. As such, the observed late apoptosis enhancement 48 h post reperfusion could therefore rather represent a mechanism to remove damaged cells associated with tissue repair. 

Third, the autophagy levels are higher when the ischemic duration increases (Figure 5). In general, these data suggest that more severe ischemia is associated with higher autophagy levels and that autophagy dynamics are dependent on the ischemia and reperfusion duration. This could explain the differences in autophagy regulation observed in the literature, as reports often focus on one or few time points post-reperfusion [23]. These differences in the literature are further complicated by the dependency of the IR-induced autophagic response on the gender and age of the model [40], as well as the differences between species. In this context, it is important to note that cell death responses to anoxia-reoxygenation differ between rodent and human cells [41], and that this will likely be reflected in the autophagy dynamics as well. Moreover, in the clinical AKI setting the ischemic time is longer than that in the model presented here. The fact that autophagy dynamics are very dependent on the species and ischemia/reperfusion time, as this study suggests, warrants the need for further evaluation in AKI patients. 

As severe ischemia is associated with decreased survival and elevated autophagy, these results bring into question whether autophagy plays an active role in renal damage or is stimulated to prevent renal IR injury. In this respect, trehalose, an autophagy inducer, seemed to have a dual effect following severe ischemia: a (slight) exacerbation of renal injury 24 h post reperfusion on the one hand and an improvement in the kidney tubular structure and reduced apoptosis 7 days following the ischemia on the other (Figure 6 and Appendix A). A similar autophagy-dependent stimulation of tissue remodeling was observed following cardiac injury, in which the beneficial effects of trehalose were blunted by the suppression of autophagy [42]. These observations, together with the lower autophagy levels associated with mild ischemia (Figure 2), suggest that autophagy may be detrimental in the acute phase of IR injury, but protective by promoting tissue repair in the recovery phase. This opens prospects for possible clinical applications of autophagy modulators such as trehalose; for example, as it functions in the repair process following injury, it could be administered peri- or post-operatively during kidney transplantation. In this respect, it is important to note that trehalose is an essential component of the “extracellular type” ET-Kyoto organ preservation solution, which is used in clinical lung transplantation [43], and which was found to be superior in the preservation of rat livers compared to the University of Wisconsin solution [44]. The molecular mechanisms of the beneficial effects of trehalose in renal IR injury, whether this is autophagy-dependent and whether this could be used in the clinical setting, require more investigation.

Due to the limitations of the rat model in this study, autophagy was only analyzed with the Western blotting and qPCR techniques. Nevertheless, it is recommended to analyze autophagy through a series of techniques [28]. Although TEM analysis revealed autophagic vesicles in the kidney tissue (Appendix A), no clear differences were observed between the Sham and I45 rats. However, TEM is performed on a very small section of the kidney, while the altered autophagy responses are likely restricted to certain renal zones (cf. positive TUNEL staining in the corticomedullary area). Similar experiments in GFP- or RFP-GFP-LC3 mice subjected to various durations of ischemia and reperfusion should gain more insight in the exact dynamics of autophagy. Nonetheless, the combination of the autophagosome marker LC3-II and autophagy degradation marker p62 analyzed in our experiments revealed the expected inverse correlation of LC3-II and p62 dynamics, indicating that these levels indeed reflected autophagy alterations in response to renal ischemia and reperfusion.

In conclusion, the differential dynamics observed in these different IR models thus partially explain the conflicting findings in the literature regarding the role of autophagy in renal IR injury, which is dependent on ischemic duration and the time of reperfusion. As such, it is important to investigate multiple time points post reperfusion and various ischemic lengths. Additionally, modulation with trehalose revealed that autophagy stimulation likely has different outcomes in tissue repair (protective) and in acute IR injury (detrimental). This information is crucial in light of the possible clinical application of (trehalose-dependent) autophagy stimulation in acute kidney injury or kidney transplantation.

## 4. Materials and Methods 

### 4.1. Ischemia-Reperfusion Injury Model

Female Sprague-Dawley rats (200–250 g; 8–10 weeks old) were housed at the KU Leuven animal facility. After at least 1 week of acclimatization, they were anesthetized by an intraperitoneal injection of 7.5 mg/kg of ketamin (Anesketin^®^, Eurovet Animal Health BV, Bladel, The Netherlands) and 2.5 mg/kg of xylazin (Xyl-M 2%^®^, Van Miert & Dams Chemie (VMD), Arendonk, Belgium). Both renal pedicles were dissected free through a midline abdominal incision and clamped “en bloc” with microaneurysm clamps to induce ischemia. Reperfusion was initiated by the removal of the clamps. Sham-operated rats underwent the same surgery without the clamping of the pedicles. Analgesics were administered daily (Vetergesic 0.1 mg/kg, CEVA, Libourne, France) following surgery. At the end of the experiment, pentobarbital (Nembutal 60 mg/kg, CEVA) was injected intraperitoneally and the rats were sacrificed by exsanguination, which allowed plasma sampling directly from the aorta. Plasma was spun down (1000× *g*; 10 min) and snap-frozen. Kidneys were collected and processed immediately, as described below. For the trehalose experiments, sterile PBS (Vehicle) or 2 g/kg bodyweight of trehalose in PBS (in a total volume of ca. 500 µL) was injected intraperitoneally 48 h and 24 h prior to surgery. The rat survival, behavior, and humane endpoints were monitored thrice daily. The animal care and experimental protocols were in accordance with the European guidelines and approved by the Ethical Committee for Animal Experimentation of KU Leuven (P053/2016).

### 4.2. Experimental Groups

First, to analyze the dynamics of autophagy during reperfusion, Sham-operated rats (Sham) and rats subjected to 45 min of ischemia (I45-mild ischemia) were divided into subgroups with various reperfusion (R) times: R0h, R1h, R3h, R6h, R24h, R48h, R7d, or R90d. Next, to assess the effects of the ischemic time, we compared the I45 mild ischemia group with rats subjected to 60 min of ischemia (severe ischemia-I60), followed by 0 h, 3 h, 24 h, and 7 days of reperfusion. Six rats were included per subgroup.

### 4.3. Kidney Function and Injury

Kidney function was assessed by plasma creatinine, measured by the kinetic Jaffé method (Hitachi/Roche Modular P, Roche Diagnostics, Diegem, Belgium) by the central laboratory of our University Hospitals. The cellular injury markers aspartate aminotransferase [26,45] (AST, colorimetric method on Hitachi/Roche Modular P) and heart-fatty acid binding protein [26,45,46] (h-FABP, enzyme-linked immunosorbant assay (ELISA)) were assessed according to the manufacturer’s instructions (HK414-Hycult Biotech, Uden, The Netherlands).

### 4.4. Transmission Electron Microscopy

Small samples of the cortex (ca. 1 mm³) were fixed in 2.5% glutaraldehyde, 0.1 M of sodium cacodylate, and 0.05% CaCl_2_ (pH 7.4) and further processed for transmission electron microscopy, as described previously [47], with minor modifications (extra staining with 1% tannic acid in veronal acetate for 1h after OsO4 postfixation). A FEI Tecnai microscope was used to examine ultrathin sections at 80–120 kV. 

### 4.5. Western Blotting

Snap-frozen kidney samples were homogenized in a radioimmunoprecipitation assay buffer (RIPA) lysis buffer containing 50 mM of Tris-HCl (pH 7.4); 150 mM of NaCl; 1 mM of EDTA; 1% Ipegal; protease, and phosphatase inhibitors (Roche, Basel, Switzerland). The protein concentration was determined through use of a Bradford protein assay (Sigma-Aldrich, Saint-Louis, MO, USA). Samples were prepared with Laemmli buffer containing β-mercaptoethanol, heated at 95 °C for 3 min and loaded on Any kD Mini-Protean TGX Precast Gel (Bio-Rad Laboratories, Hercules, CA, USA). SDS-PAGE was performed with a constant voltage of 150 V. Next, the proteins were blotted on polyvinylidene difluoride (PVDF) membranes using the semi-dry Trans-Blot Turbo Transfer system (Bio-Rad Laboratories). Membranes were blocked for 1h at room temperature with PBS-Tween (0.1%) containing 5% milk powder (MP), followed by incubation with the primary antibody diluted in PBS-T and 2% MP overnight at 4 °C. Next day, the membranes were washed 3 times with PBS-T and then incubated with the secondary horseradish peroxidase (HRP)-coupled antibody for 45 min with PBS-T + 2% MP. After washing three times with PBS-T, immunoreactive bands were visualized through enhanced chemoluminescence (Pierce ECL Western Blotting Substrate, Thermo Fisher Scientific, Waltham, MA, USA), followed by detection and band intensity quantification using the Chemidoc MP technology and associated Imagelab software (Bio-Rad Laboratories). 

### 4.6. Antibodies 

The following antibodies and reagents were used for the Western blotting experiments. Anti-LC3 (5F10, Nanotools), anti-S6 (2217, Cell Signaling Technology, Danvers, MA, USA), anti-phospho-S6 (4858, Cell Signaling Technology), anti-cleaved caspase 3 (9664, Cell Signaling Technology), anti-SQSTM1/p62 (P0067, Sigma-Aldrich, Saint-Louis, MO, USA), anti-Bcl-2 (sc-492, Santa Cruz Biotechnology, Dallas, TX, USA), and anti-Bax (sc-493, Santa Cruz Biotechnology). Anti-GAPDH (G8715, Sigma-Aldrich) served as an internal control. Secondary antibodies are HRP-linked anti-mouse IgG (7076, Cell Signaling Technology) and HRP-linked anti-rabbit IgG (1706515, Bio-Rad Laboratories, Hercules, CA, USA).

### 4.7. Quantitative Real-Time Polymerase Chain Reaction

RNA was extracted from kidney tissues using Trizol reagent and chloroform, followed by an additional purification step with the RNeasy mini kit (Qiagen, Hilden, Germany), according to the manufacturer’s protocol. The mRNA expression levels in kidney tissues were analyzed using quantitative real-time polymerase chain reaction (Q-RT-PCR). Reverse transcription was performed at 37 °C for 1 h with the M-MLV Reverse Transcriptase along with FSBuffer and Rnase out (Life Technologies, Carlsbad, CA, USA). Q-RT-PCR was performed with the PCR mastermix of Applied Biosystems (1.503.193, Foster City, CA, USA). Thermal cycling conditions were composed of cDNA initially denatured at 95 °C for 60 s, and then amplified by PCR for 45 cycles (95 °C for 5 s, 60 °C for 30 s). Experiments were carried out in duplicates. Using the 2-ΔΔCt method [48], the relative quantification in gene expression was determined. Data are expressed as the relative differences (fold change) between the sham and IR samples after correction for GAPDH expression. 

### 4.8. qPCR Taqman Probes

The following reagents from TaqMan Gene Expression Assays (Applied biosystems, Foster Cyti, CA, USA) were used for RT-PCR experiments: GAPDH (Rn01775763_g1), BECN1 (Rn00586976_m1), LC3 (Rn02132764_S1), p62 (Rn00709977_m1), Bcl-2 (Rn99999125_m1), Bax (Rn01480160_g1), Bim (Rn00674175_m1), Hsp70 (Rn00583013_S1), ICAM-1 (Rn00564227_m1), IL-10 (Rn00563409_m1), and IL-6 (Rn01410330_m1).

### 4.9. Terminal Deoxynucleotidyl Transferase End Labeling

For the detection of oligonucleosomal DNA cleavage via terminal deoxynucleotidyl transferase end labeling (TUNEL), tissue sections were deparaffinized in toluene (2 × 5 min), rehydrated in distilled water (5 min), and pretreated with 3% citric acid (60 min) to remove tissue calcification. Endogenous peroxidase was quenched by incubating sections for 15 min in 0.9% hydrogen peroxide. Thereafter, TUNEL was performed using an ApopTag Plus Peroxidase In Situ Apoptosis Detection Kit (Merck-Millipore, Burlington, MA, USA) according to the instructions of the manufacturer. Slides were observed with an Olympus BX61 microscope at 20× magnification, and pictures were taken with the Olympus Stream Essentials 1.9 software. Quantification of the TUNEL staining was performed with the “color threshold” function of ImageJ, using the color space RGB and the same parameters for each analyzed picture.

### 4.10. Statistical Analysis

If data were normally distributed, unpaired t-tests were used for the comparison of 2 groups (Sham versus IR group) or a one-way ANOVA for 3 groups (Sham, I45, and I60) with Tukey Multiple Comparison test post-hoc. F-test was performed to compare variances. If the variances were significantly different, Welch’s correction was applied. If a group was not normally distributed, non-parametric tests (Mann–Whitney or Kruskal–Wallis test with Dunns post-hoc) were performed. For relative Western blotting and qPCR data, values were normalized to the mean of the control Sham group. Outliers were removed based on the Grubb’s test. Normal distribution was analyzed with the Kolmogorov–Smirnov test (for small sample sizes). 

## Figures and Tables

**Figure 1 ijms-21-07185-f001:**
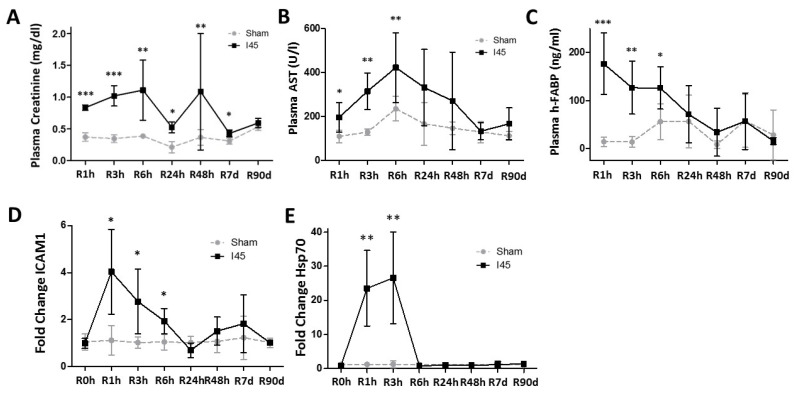
Kidney function is reduced, injury is increased, and tissue inflammatory processes are activated after mild ischemia and reperfusion. Rats, either Sham-operated (Sham) or subjected to 45 min of renal ischemia (I45), were sacrificed at various time points post reperfusion (R0h, R1h, R3h, R6h, R24h, R48h, R7d, and R90d). Plasma was collected to measure the creatinine (**A**), aspartate aminotransferase (AST) (**B**), and heart-fatty acid binding protein (h-FABP) (**C**). Kidneys were collected and analyzed for the mRNA expression of intracellular adhesion molecule-1 (ICAM-1) (**D**) and heat shock protein 70 (Hsp70) (**E**). * *p* <0.05, ** *p* < 0.01, *** *p* < 0.001. *N* = 6.

**Figure 2 ijms-21-07185-f002:**
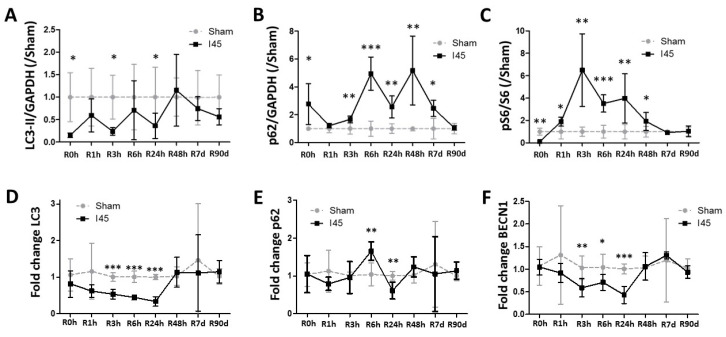
Autophagy is suppressed post reperfusion following mild ischemia. Rats, either Sham-operated (Sham) or subjected to 45 min of renal ischemia (I45), were sacrificed at various time points post reperfusion (R0h, R1h, R3h, R6h, R24h, R48h, R7d, and R90d). Kidney tissue was collected and analyzed by Western blotting for LC3 (**A**), p62 (**B**), S6, and phosphorylated S6 (pS6) (**C**), and by qPCR for the mRNA expression of LC3 (**D**), p62 (**E**), and BECN1 (**F**). * *p* <0.05, ** *p* < 0.01, *** *p* < 0.001. *N* = 6.

**Figure 3 ijms-21-07185-f003:**
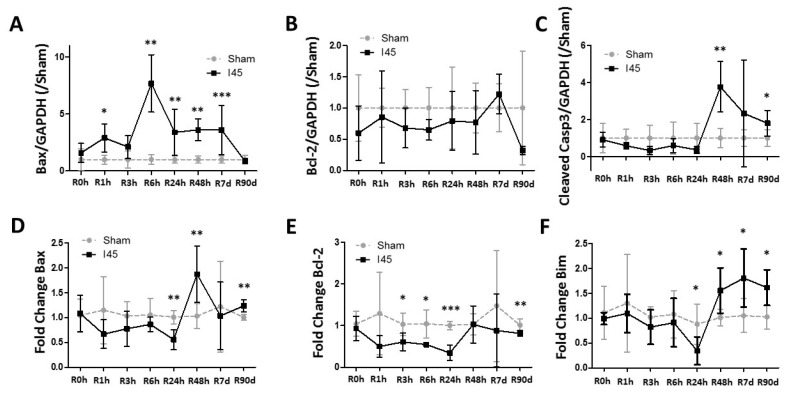
Apoptosis increases post reperfusion following mild ischemia. Rats, either Sham-operated (Sham) or subjected to 45 min of renal ischemia (I45), were sacrificed at various time points post reperfusion (R0h, R1h, R3h, R6h, R24h, R48h, R7d, and R90d). Kidney tissue was collected and analyzed by Western blotting for Bax (**A**), Bcl-2 (**B**), and Cleaved Caspase 3 (**C**), and by qPCR for the mRNA expression of Bax (**D**), Bcl-2 (**E**), and Bim (**F**). * *p* <0.05, ** *p* < 0.01, *** *p* < 0.001. *N* = 6.

**Figure 4 ijms-21-07185-f004:**
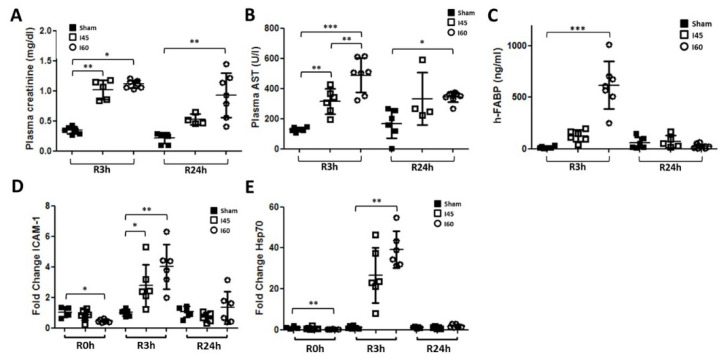
Kidney function is reduced and injury is increased following severe compared to mild or no ischemia. Rats, either Sham-operated (Sham) or subjected to 45 min (I45) or 60 min (I60) of renal ischemia, were sacrificed at various time points post reperfusion (R0h, R3h, and R24h). Plasma was collected to measure the creatinine (**A**), AST (**B**), and h-FABP (**C**). Kidneys were collected and analyzed for the mRNA expression of ICAM-1 (**D**) and Hsp70 (**E**). * *p* <0.05, ** *p* < 0.01, *** *p* < 0.001. *N* = 6.

**Figure 5 ijms-21-07185-f005:**
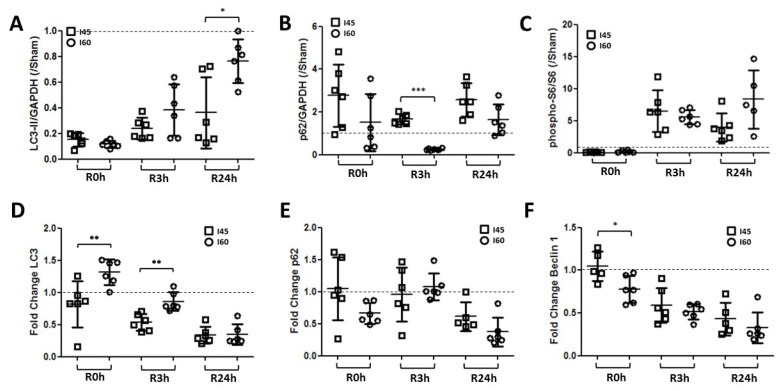
Autophagy is increased during ischemia and after reperfusion following severe ischemia compared to mild ischemia. Rats, either Sham-operated (Sham) or subjected to 45 min (I45) or 60 min (I60) of renal ischemia, were sacrificed at various time points post reperfusion (R0h, R3h, and R24h). Kidneys were collected and analyzed by Western blotting and qPCR for LC3 (**A**), p62 (**B**), S6, and phosphorylated S6 (pS6) (**C**), and by qPCR for the mRNA expression of LC3 (**D**), p62 (**E**), and BECN1 (**F**).The relative change in I45 and I60 compared to the corresponding Sham group (represented by the dashed line at y = 1) was plotted. * *p* <0.05, ** *p* < 0.01, *** *p* < 0.001. *N* = 6.

**Figure 6 ijms-21-07185-f006:**
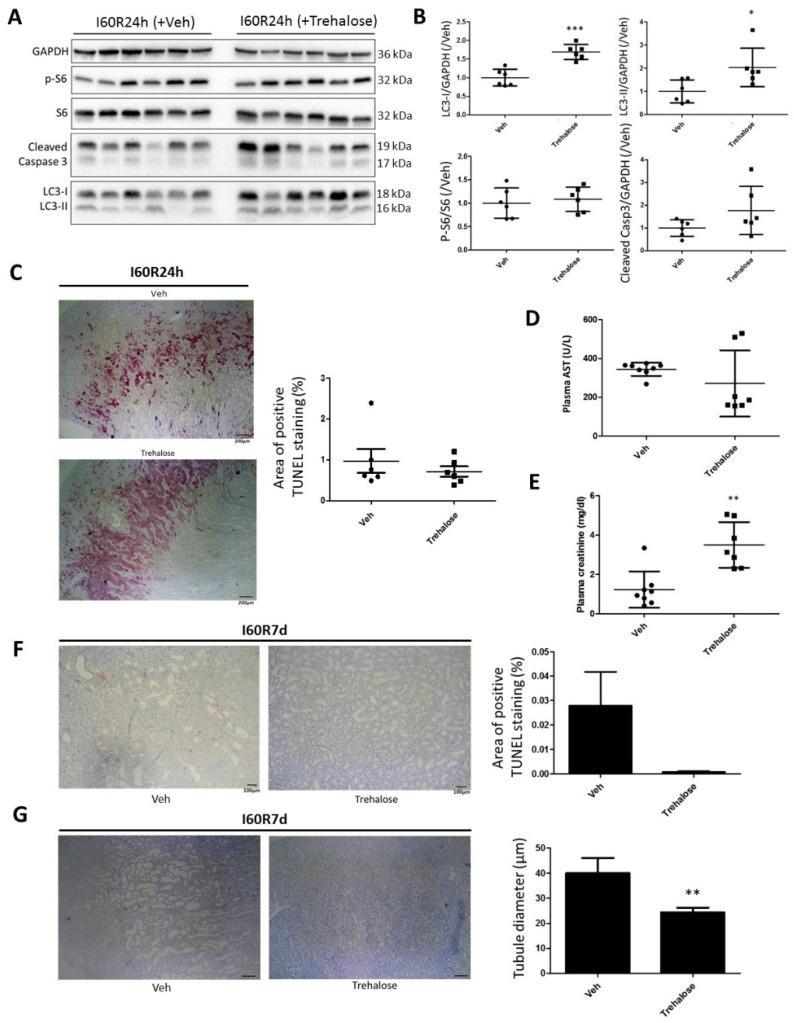
Trehalose induces autophagy and improves the renal structure post reperfusion following severe ischemia. Rats were injected with vehicle (Veh) or trehalose 48 h and 24 h prior to 60 min of renal ischemia (I60), and sacrificed 24 h (I60R24h) or 7 days (I60R7d) post reperfusion. Kidneys of I60R24h were analyzed by Western blotting for the annotated markers (**A**) and compared between Veh- and trehalose-treated for LC3-I, LC3-II, phosphorylated S6 (pS6), and cleaved caspase 3 (**B**). Kidney sections of I60R24h were also stained with TUNEL (**C**) and the plasma levels of AST (**D**) and creatinine (**E**) were determined. In the kidney sections of I60R7d, TUNEL staining was performed (**F**) and the average tubule diameter was assessed (**G**). * *p* < 0.05, ** *p* < 0.01, *** *p* < 0.001. *N* = 6.

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
