# Peer review of "Autophagy Dynamics and Modulation in a Rat Model of Renal Ischemia-Reperfusion Injury"

_ijms, 2020, doi:10.3390/ijms21197185_

Round 1

Reviewer 1 Report

Decuypere et al. report the role of autophagy in an IRI rat model. The manuscript is well written and of scientific soundness.

Could the authors summarise on a table the mechanisms by which autophagy works and its involvement in the necrosis and apoptosis settings?

Author Response

Reviewer 1:

Decuypere et al. report the role of autophagy in an IRI rat model. The manuscript is well written and of scientific soundness.

We thank the reviewer for his/her compliments on our manuscript and for the comment.

Could the authors summarise on a table the mechanisms by which autophagy works and its involvement in the necrosis and apoptosis settings?

We agree with the reviewer that the introduction is rather brief regarding the mechanism of autophagy, and how it is negatively and positively associated with cell death pathways. However, we feel that a (comprehensive) description of autophagy and its link with apoptosis and necrosis in a table would be out of the scope of this paper. To address this comment, we have therefore added more information in the introduction on the molecular mechanism of autophagy and shortly on how it could prevent or assist cell death. In addition, we have added several references to reviews discussing this matter. Please see p. 2 lines 44-51 and lines 56-58. We hope that we have hereby addressed your comment in a satisfactory manner.

Reviewer 2 Report

The subject of the study is interesting, the authors performed a lot of experimental work, and generally, the manuscript is well written. I think that this well-conducted and laborious study deserves publication.

I have the following comments:

  1. To evaluate the activation status of mTOR, the authors assessed the level of phosphorylation of ribosomal protein S6. However, rpS6 is phosphorylated by many kinases, and most of them do not belong directly to the mTORC1 pathway (Biever, DOI: 10.3389/fnmol.2015.00075). Thus, the data form the study cannot support the authors’ statement about the role of mTOR. I suggest the authors assessing closer to mTOR targets, such as phosphorylated p70S6 kinase or phosphorylated 4E-BP1 or omitting the related statements.
  2. The fact that severe I-R injury did not enhance apoptosis is surprising since many studies implicate this type of cell death in similar experimental models of kidney I-R injury. A discussion of this point would be welcome.
  3. Since in 60 min I-R injury renal damage is more severe, how do the related results about apoptosis can provide an explanation? Do other types of cell death, not evaluated in the study, play a role? For instance, in primary human and mouse renal proximal tubular epithelial cells, ferroptosis is responsible for reoxygenation-induced cell death, while apoptosis takes place during the anoxic phase of I-R injury (Eleftheriadis, DOI: 10.3390/biology7040048). Does the absence of apoptosis in R0 was due to the relatively short time of ischemia? Also, in the in vivo condition, where inflammation takes place, necroptosis may play a significant role as well (Linkermann, DOI: 10.1681/ASN.2014030262). A discussion would be welcome.
  4. I agree with the experimental design of the study. However, since, eventually, such studies aim to advance medicine, a comment about the short time of ischemia used in experimental models, compared to the considerably longer time of ischemia encountered in the clinic, would be useful.
  5. Also, the interspecies differences should be commented. For instance, primary human renal proximal epithelial cells are much more vulnerable to anoxia than mouse cells (death after 4 hours vs. 48 hours) but resist somewhat more to reoxygenation (Eleftheriadis, DOI: 10.3390/biology7040048).
  6. Finally, because the study includes many results, I suggest the authors resuming the findings within a short sentence at the end of each result subsection.

Author Response

Reviewer 2:

The subject of the study is interesting, the authors performed a lot of experimental work, and generally, the manuscript is well written. I think that this well-conducted and laborious study deserves publication.

We thank the reviewer for his/her compliments on our manuscript and the insightful comments. Please find a point-by-point response to the comments below.

I have the following comments:

  1. To evaluate the activation status of mTOR, the authors assessed the level of phosphorylation of ribosomal protein S6. However, rpS6 is phosphorylated by many kinases, and most of them do not belong directly to the mTORC1 pathway (Biever, DOI: 10.3389/fnmol.2015.00075). Thus, the data form the study cannot support the authors’ statement about the role of mTOR. I suggest the authors assessing closer to mTOR targets, such as phosphorylated p70S6 kinase or phosphorylated 4E-BP1 or omitting the related statements.

Indeed, we agree with the reviewer that rpS6 is only indirectly phosphorylated by mTOR through p70S6K and that rpS6 is phosphorylated by other kinases as well. Moreover, to link the these data with autophagy, phosphorylated ULK1 levels would be the best assessment. We have therefore, as requested by the reviewer, altered and nuanced the statements on mTOR activity and included the proposed reference, please see p. 3 line 108-110, lines 114-117, p. 6 line 176, p.7 lines 200-201 and p.9 lines 246-251.

  1. The fact that severe I-R injury did not enhance apoptosis is surprising since many studies implicate this type of cell death in similar experimental models of kidney I-R injury. A discussion of this point would be welcome.
  2. Since in 60 min I-R injury renal damage is more severe, how do the related results about apoptosis can provide an explanation? Do other types of cell death, not evaluated in the study, play a role? For instance, in primary human and mouse renal proximal tubular epithelial cells, ferroptosis is responsible for reoxygenation-induced cell death, while apoptosis takes place during the anoxic phase of I-R injury (Eleftheriadis, DOI: 10.3390/biology7040048). Does the absence of apoptosis in R0 was due to the relatively short time of ischemia? Also, in the in vivo condition, where inflammation takes place, necroptosis may play a significant role as well (Linkermann, DOI: 10.1681/ASN.2014030262). A discussion would be welcome.

Since these two comments are related, we would like to address them together. Indeed, we agree with the reviewer that it is surprising there was no immediate increase in apoptosis in our model. However, enhanced caspase 3 cleavage and Bim levels were observed after 48 h of reperfusion. These data therefore suggest that the initial (acute) injury post-reperfusion is likely more associated with other (inflammatory) types of cell death (e.g. necrosis, caspase3-independent pyrroptosis, ferroptosis or necroptosis). Indeed, in previous work, we have shown that necroptosis inhibitor Nec-1 reduces the positive TUNEL staining 3 h post-reperfusion in our model (Decuypere et al. Transplantation 2017 101(11):e330-e336), suggesting necroptosis is activated shortly following reperfusion. However, since Nec-1 did not affect injury markers AST, h-FABP and plasma creatinine levels, other types of cell death are likely activated as well. As such, the observed late apoptosis enhancement 48 h post-reperfusion could therefore rather represent a mechanism to remove damaged cells associated with tissue repair. We have added several lines in the discussion on the involvement of different types of cell death in this model, see p. 9, lines 258-266.

  1. I agree with the experimental design of the study. However, since, eventually, such studies aim to advance medicine, a comment about the short time of ischemia used in experimental models, compared to the considerably longer time of ischemia encountered in the clinic, would be useful.

We agree with the reviewer that the model studied here is subjected to short ischemia times. As such, it is difficult to extrapolate the autophagy dynamics observed in this model towards the clinical setting. However, in regard to the findings of trehalose, and the assumption that trehalose would aid in the repair mechanisms in the recovery phase post-reperfusion (possibly through autophagy stimulation), its possible clinical use is less dependent on ischemia time and still relevant in the clinical setting.

We have addressed this on p. 10 line 275-278 and line 295-296

  1. Also, the interspecies differences should be commented. For instance, primary human renal proximal epithelial cells are much more vulnerable to anoxia than mouse cells (death after 4 hours vs. 48 hours) but resist somewhat more to reoxygenation (Eleftheriadis, DOI: 10.3390/biology7040048).

Indeed, this is an important remark and we have addressed this on p. 10 lines 272-275.

  1. Finally, because the study includes many results, I suggest the authors resuming the findings within a short sentence at the end of each result subsection.

We have added a sentence at the end of each result section to summarize the main findings.

Round 2

Reviewer 2 Report

The authors address the raised issues.